# Genetic Variation and Metapopulation Structure Inform Recovery Goals in a Threatened Species

**DOI:** 10.3390/genes16060694

**Published:** 2025-06-08

**Authors:** Molly J. Garrett, Courtney J. Conway, Lisette P. Waits, Paul A. Hohenlohe

**Affiliations:** 1Department of Fish and Wildlife Sciences, College of Natural Resources, University of Idaho, Moscow, ID 83844, USA; mollyjgarrett@gmail.com (M.J.G.); lwaits@uidaho.edu (L.P.W.); 2U.S. Geological Survey, Idaho Cooperative Fish & Wildlife Research Unit, University of Idaho, Moscow, ID 83844, USA; cconway@uidaho.edu; 3Department of Biological Sciences, College of Science, University of Idaho, Moscow, ID 83844, USA

**Keywords:** conservation units, effective population size, ground squirrels, GT-seq, *Urocitellus*

## Abstract

Background: Monitoring genetic parameters is important for setting effective conservation and management strategies, particularly for small, fragmented, and isolated populations. Small, isolated populations face increased rates of genetic drift and inbreeding, which increase extinction risk especially when gene flow is limited. Methods: Here, we applied a Genotyping-in-Thousands by sequencing (GT-seq) panel to inform recovery action for the federally threatened northern Idaho ground squirrel (*Urocitellus brunneus*). We evaluated genetic diversity, structure, connectivity, and effective population size to address species recovery goals. Results: We delineated three types of conservation units: (1) three evolutionarily significant units that represent long-term population structure and variation, (2) nine management units that reflect current demographic connectivity and restrictions to gene flow, and (3) three adaptive units that capture adaptive differentiation across the species range. Effective population sizes per management unit were small overall (mean 38.16, range 2.3–220.9), indicating that recovery goals of 10 subpopulations with *N*_e_ > 500 have not been reached. Conclusions: Our results support the maintenance of connectivity within evolutionarily significant units through the restoration of dispersal corridors. Next steps could include further sampling of some subpopulations with low sample sizes, unsampled subpopulations, and subpopulations that are geographically isolated. Genotyping future samples with the same GT-seq panel would help to detect dispersal, assess effective population size, monitor the effects of inbreeding, and evaluate adaptive differentiation to monitor the effects of management action and environmental change.

## 1. Introduction

Understanding how environmental factors influence demography, connectivity, local adaptation, and ultimately resilience of small populations is critical to set conservation strategies and implement effective habitat management plans [1,2]. For example, increased levels of connectivity can facilitate gene flow between small populations that otherwise are vulnerable to high levels of genetic drift, while high levels of connectivity can also create genetically homogenous populations that may lack locally adapted phenotypes. Decreased connectivity may lead to inbreeding depression, where individuals suffer reduced fitness due to inbreeding [3,4]. Inbreeding depression can decrease the viability of small populations, especially in combination with underlying stochastic environmental, demographic, and genetic processes that are more pronounced in small populations [5,6]. Therefore, examining the underlying genetic factors is critical for effectively managing rare species.

An important metric for evaluating the relative importance of genetic factors for management and conservation is effective population size (*N_e_*), which determines the expected rate of loss of genetic variation by genetic drift. Effective population size can also be used to estimate levels of inbreeding, which can affect fitness and viability of small populations [7]. Contemporary *N_e_* estimates reflect evolutionary processes in one or a few recent generations and provide insight for conservation of populations facing ongoing threats [7]. Estimates of *N_e_* can provide metrics for monitoring, goals for population recovery, and information to guide actions for threatened, fragmented, or small populations.

Determining *N_e_* and other population genetic parameters is especially valuable for the conservation of species that exhibit a metapopulation structure. A metapopulation is an assemblage of interconnected breeding populations that occur in semi-independent subpopulations [8,9,10]. The interactions between genetic drift, gene flow, and local adaptation are variable between highly connected or isolated subpopulations. For example, small, isolated subpopulations can have relatively small *N_e_*, a high degree of inbreeding, and high risk of local extinction [11,12]. On the other hand, metapopulations that are well connected can maintain genetic variation, with gene flow overcoming the loss of variation within subpopulations while still maintaining local adaptation [13,14]. As habitat fragmentation increases, monitoring gene flow and local adaptation within metapopulations is important for assessing population persistence for conservation and for designing management actions like genetic rescue or translocations.

Advancements in genomic sequencing technologies have provided increased resolution and aided the identification of adaptive loci, which have improved our ability to monitor and manage wildlife populations against the effects of environmental change and biodiversity loss [15,16]. Additionally, sequencing advances allow more neutral loci to be sequenced across the genome, which provides more accurate estimates of *N_e_* and other metrics estimated with neutral loci (e.g., gene flow and connectivity). One such advance in amplicon sequencing is Genotyping-in-Thousands by sequencing (GT-seq) [17]. GT-seq targets a pre-identified set of genetic loci that are informative for a specific suite of objectives, and many samples can be genotyped to monitor genetic change over time and/or space. In addition to its relative low costs, GT-seq can be applied to lower-quality DNA samples relative to some other genomic techniques, so it is applicable to minimally invasive sampling in rare or threatened populations [18,19,20]. Compared to other genotyping techniques like microsatellites, GT-seq has a streamlined bioinformatics processing pipeline that allows for reproducibility between laboratories and includes the analysis of adaptive loci.

Genomic data can contribute to the delineation of conservation units within species and thus aid legal classifications and management efforts [21,22,23,24]. The evolutionarily significant unit (ESU) is the broadest conservation unit category under the species level. The ESU, as defined by Waples [25], classifies populations based on the degree of reproductive isolation and evolutionary potential (i.e., adaptive capacity). Notably, this definition of the ESU is the same as that of the distinct population segment (DPS) under the U.S. Endangered Species Act. ESUs represent large, intraspecific units that have undergone some degree of historical isolation, and likely have important adaptive differences between units [26]. Within the ESU, management units (MU) delineate populations that are demographically independent with restricted gene flow [27]. Unlike ESUs which are concerned with historical population structure and evolutionary history, MUs define groups based on current demographic independence and the degree of connectivity. Therefore, MUs are intended for setting and assessing short-term management goals (e.g., genetic or demographic monitoring) [28]. Adaptive units (AUs) further differentiate groups based on adaptive variation [21,26,29]. Incorporating variation at outlier or putatively adaptive loci for AUs can illuminate patterns of adaptive differentiation to prioritize resources to those populations in need of immediate management action, such as translocations or assisted migration [30,31].

Northern Idaho ground squirrels (*Urocitellus brunneus*, hereafter NIDGS) are endemic to a 1600 km^2^ area in western Idaho, U.S.A., and are federally listed as a threatened species [32] with an estimated population size of 2000–3000 individuals [33]. NIDGS are obligate hibernators, spending 8–10 months hibernating each year [34,35]. NIDGS mate in the spring soon after females emerge from hibernacula in late March-April, reproduce only once per year, produce a litter of 2–7 pups, and immerge back into hibernacula in July–August. The maximum life-span of free-ranging NIDGS recorded is six years for males and eight years for females, but it is rare for NIDGS to live longer than three years for males and four years for females. Females often begin breeding during the first year after birth (as yearlings), and males often wait until their second season to initiate breeding. Hence, their generation length is approximately two-three years [32,33,34,35].

NIDGS are semi-colonial, occurring in distinct sites in xeric meadows and small rocky openings interspersed within a matrix of non-suitable habitat (such as coniferous forest). Dispersal of individuals between colonies seems to be relatively rare. Individuals rarely disperse farther than 200 m. Minimal dispersal events have been documented in NIDGS >500 m, and no movements have been documented from mark-recapture studies of >1.2 km from their capture location [36]. NIDGS are federally listed as threatened, and persist within only a small fraction (<1%) of their former range [37]. Numerous causes have been proposed to explain their rarity, such as anthropogenic fire suppression which results in forest encroachment that eliminates NIDGS habitat [32,38,39,40], competition for food or burrowing space with the larger, sympatric Columbian ground squirrel (*U. columbianus*) [40,41], reduction in dispersal corridors among sites [36,42,43], reduced survival due to sylvatic plague [44,45], and negative effects of climate change [35].

In 2003, the United States Fish and Wildlife Service (USFWS) produced a NIDGS recovery plan that delineated actions and objectives to recover and protect the species [32]. According to the plan, delisting of the NIDGS from the Endangered Species Act can occur when at least 10 subpopulations each maintain an average *N_e_* of greater than 500 individuals over five consecutive years. This estimate is based on a historical population estimate of 5000 individuals [32], which managers have used to base their recovery criteria of NIDGS having a total metapopulation *N_e_* of greater than 5000 individuals. Other objectives of the recovery plan include defining metapopulation structure and conservation units for the NIDGS more accurately [26], as well as understanding connectivity among subpopulations.

Samples from extant populations of NIDGS have been collected and analyzed over the past 20 years using various DNA sequencing and genotyping techniques, from allozymes to microsatellites to restriction site associated DNA sequencing (RADseq) [36,42,43,46,47]. Due to the threatened status of the species, regulatory agencies currently require less stressful, minimally invasive genetic sampling techniques including plucking hair and taking buccal swabs. Prior studies on the species’ population genetics have focused on evaluating phylogenetic relationships, genetic diversity, and gene flow [36,42,43,47]. Barbosa et al. [42] highlighted the importance of assessing NIDGS genetic parameters with both neutral and adaptive loci, since small, isolated populations may be reliant on locally adapted genotypes for population persistence. However, Barbosa et al. [42] had small sample sizes for some sites because many of the minimally invasive samples did not produce sufficient DNA quality or quantity for RADseq. Therefore, we were motivated to develop a GT-seq panel that would allow us to genotype previous, current, and future samples to obtain standardized assessments of genetic diversity and gene flow among extant populations, as well as adaptive differentiation among populations. We selected neutral and putatively adaptive SNPs identified in Barbosa et al. [42] to create the NIDGS GT-seq panel [48].

To aid in species recovery efforts, we applied the GT-seq SNP panel of 305 loci [48] to evaluate NIDGS genetic diversity, genetic structure, connectivity, and effective population size, and to define conservation units across 18 sampling areas (sites). We classified subpopulations into three types of conservation units: (1) ESUs to capture long-term population structure and variation, (2) MUs to reflect current demographic connectivity and restrictions to gene flow, and (3) AUs to capture adaptive differentiation across the species’ range [21,23,26,29]. We estimated *N_e_* within and connectivity among MUs to directly address the criteria necessary to evaluate recovery objectives required for delisting the species. This study represents the most extensive genetic analysis of this threatened species to date and provides critical information for addressing several of the listed recovery goals.

## 2. Materials and Methods

### 2.1. Study Area and Sample Collection

To capture individuals, Tomahawk live traps (Tomahawk Live Trap Co., Hazelhurst, WI, USA; 13 × 13 × 41 cm and 15 × 15 × 50 cm) were baited and placed near burrows or logs at regularly spaced intervals within occupied sites in 2016 and 2020 [42,43,47] (Figure 1, Table 1). All study sites were located in Adams and Valley Counties, Idaho, spread throughout the extant range of the NIDGS. Study site elevations ranged from 1060 m to 1700 m. The mean annual precipitation for the past 10 years at the study sites was 715 mm, most of which fell as winter snow. Study sites were within a landscape that included coniferous forest interspersed with riparian corridors, rocky scabs, and meadows. Ponderosa pine (*Pinus ponderosa*) and Douglas fir (*Pseudotsuga menziesii*) dominated the overstory in forested areas at our study sites. Common forest understory plants included snowberry (*Symphoricarpos* spp.), spiraea (*Spiraea* spp.), pine grass (*Calamagrostis rubescens*), elk sedge (*Carex geyeri*), heart-leaf arnica (*Arnica cordifolia*), and wild strawberry (*Fragaria* spp.). Common plants in xeric meadows included big sagebrush (*Artemisia tridentata*), bluegrass (*Poa* spp.), bromes (*Bromus* spp.), buckwheat (*Eriogonum* spp.), wild onion (*Allium* spp.), and biscuitroot *(Lomatium* spp.). Cattle grazing, hunting, and camping were common human land uses, though human presence on our study sites was rare.

We swabbed the inside of each captured squirrel’s cheek with a sterile cotton swab (Lakewood Biomedical, Dallas, TX, USA) to collect epithelial cells, which we repeated five times per individual. All replicate buccal swabs per individual were preserved in the same tube in ATL buffer (QIAGEN, Inc., Venlo, Netherlands) until DNA extraction. All NIDGS were trapped and handled following protocols and procedures approved by the University of Idaho Animal Care and Use Committee (IACUC #2015-53 and #2019-28). Our activities were approved under an Idaho Department of Fish and Game scientific collecting permit (SCP #120629) and a U.S. Fish and Wildlife Service recovery permit (TE94776A-3).

### 2.2. GT-seq Library Preparation, Sequencing, and Genotyping

We used Qiagen Blood and Tissue extraction kits (QIAGEN, Inc.) to extract DNA from buccal swab samples using standard protocols. The buccal swab samples were extracted in a low-quantity DNA facility with no PCR products or high quantity DNA sources.

Detailed steps of the GT-seq panel SNP selection, primer development, optimization, library preparation, and genotyping are described in Garrett et al. [48]. We sequenced seven GT-seq libraries of buccal swab extracts using a single-read 118 bp cycle on Illumina NovaSeq SP at the University of Oregon Genomics and Cell Characterization Core Facility (G3CF) and a paired-end 75 bp cycle on Illumina NextSeq at Seqmatic LLC (San Francisco, CA, USA). We used the GT-seq pipeline v. 3 on GitHub (https://github.com/GTseq, accessed on 4 June 2024) for demultiplexing and genotyping, which was chosen after comparison among genotyping methods and validation against RAD sequencing data [47]. After testing different allele ratios for calling heterozygotes, we modified the GTseq_Genoytper_v3.pl script to call heterozygotes with an allele ratio between 0.4 and 2.5, from the original 0.2–2.0 range. Using the R package ADEGENET v2.1.10 [49], individuals were removed with missing values greater than 25% and loci were removed with missing values greater than 50%. Monomorphic loci were also removed. Additionally, loci with a per locus *F_IS_* value less than −0.15 were also removed.

For each site per year, we used the basic.stats function from HIERFSTAT R package v0.5-11 [50] to estimate observed and expected heterozygosity. From the private_alleles function from POPPR R package v2.9.4 [51], we quantified private alleles per site per year.

We estimated the isolation by distance (IBD) using the mantel function in the VEGAN R package v2.6-4 [52] for both 2016 and 2020 separately. We estimated IBD with two metrics of genetic distance with all loci, as well as neutral and putatively adaptive loci independently, for 2016 and 2020. We used the as.dist function in HIERFSTAT R package v0.5-11 [50] to calculate pairwise *F_ST_* and the dist.genpop function in ADEGENET R v2.1.10 package [49] to calculate Nei’s genetic distance. We compared genetic distances for each site with ≥6 samples against pairwise Euclidean distances for all sites. We calculated IBD correlation coefficients using both Pearson (with and without the log transformation of Euclidean distance) and Spearman’s rank correlation. Below, we describe IBD results for *F_ST_* and Spearman’s rank correlation, but all are included in the Appendix A.

### 2.3. Evolutionarily Significant Units

We estimated reproductive isolation and evolutionary potential through exploring population structure and genetic differentiation by using neutral and putatively adaptive SNPs. First, we performed principal component analysis (PCA) to visually inspect population structure. We used the dudi.pca function in ADE4 R package v5.7-1 [53] then the ggplot function in GGPLOT2 v3.5.0 package to create the PCA plots [54]. Next, we used the Bayesian clustering analysis as implemented in program STRUCTURE v2.3.4 [55]. We applied an admixture model with a burn-in period of 100,000 and a run length of 500,000 MCMC replicates to determine the number of genetic clusters (*K*) from 1 to 20 over 10 iterations each. We identified the optimal *K* for each dataset according to the Evanno method [56] and the rate of change in the likelihood values [55]. We visualized STRUCTURE outputs with bar-plots and geographic pie charts using the following R script (https://github.com/sakura81/PYRA_Genomics, accessed on 4 June 2024) and R package SCATTERPIE v0.2.1 [57]. We also quantified private alleles per ESU once they were defined.

### 2.4. Management Units

We investigated current population structure with the same PCA and STRUCTURE techniques described above using only neutral SNPs. We calculated pairwise *F_ST_* with only neutral loci with function boot.ppfst in HIERFSTAT R package v0.5-11 [50] with 999 bootstraps to estimate genetic distinctiveness.

We used the program Population Graphs as implemented in the POPGRAPHS R package v1.5.3 [58] to assess gene flow and connectivity among sites. The program uses a graph theory approach to describe genetic covariance among sites (i.e., nodes), where connections (i.e., edges) between sites represent genetic exchange. Edges between nodes may represent dispersal corridors or connecting habitat patches, or more aspatial theoretical connections, but are an important indicator that gene flow between nodes is possible if gene flow is not actively occurring [59,60,61]. We visualized gene flow using a cutoff of 0.90 [23]. We calculated two measures of connectivity between nodes: betweenness, the number of times that a node acts as a bridge along the shortest path between two other nodes, and degree, the total number of edges connected to a particular node in a graph [60,62,63].

After we delineated MUs, we estimated *N_e_* for each MU using the linkage disequilibrium method of NeEstimator v2.01 [64] for 2016 and 2020 separately. We estimated *N_e_* with the linkage disequilibrium method as we did not sample the same individuals over multiple time periods. We only included neutral SNPs and excluded alleles with frequencies less than a critical value of 0.05. Additionally, observed heterozygosity was calculated for each MU.

### 2.5. Adaptive Units

We quantified adaptive differentiation with PCA and STRUCTURE as described above but using only adaptive SNPs. For 2016 and 2020 separately, we tested for private alleles in candidate AU arrangements of *K* = 3, 4, and 5 to examine how adaptive allelic variation is spread across the species range at different grouping levels.

### 2.6. The Spatially Disjunct Population of Round Valley

While most NIDGS subpopulations occur in a 29 km × 37 km area in Adams County, the southernmost population, Round Valley (RV), is ~70 km from all other populations and is restricted to a 3 km × 4 km area in Valley County (Figure 1). We obtained one tissue sample from Round Valley in 2022 and sequenced it with the NIDGS GT-seq panel. We included this one sample in PCA with all the other buccal swab samples and quantified private alleles for Round Valley as compared to all other sites. We acknowledge that we cannot draw any statistical conclusions with just one sample, but the sample can provide preliminary information on how divergent Round Valley may be compared to the other populations.

## 3. Results

### 3.1. Genotyping

Sequencing yielded 201,904 mean reads per individual (range = 10–18,802,287) and 116,184 mean on-target reads per individual (range = 1–11,219,616). After filtering, the final dataset included 182 neutral and 70 putatively adaptive SNPs (total = 252 SNPs). Two samples were removed due to possible laboratory contamination or mislabeling/mishandling of samples. For 2016, there were 201 individuals from 14 sites with a mean sample size per site of 14. For 2020, there were 207 individuals from 18 sites with a mean sample size per site of 18 (Table 1).

For all sites except for Rocky Top (RT), observed heterozygosity was highest with neutral SNPs only, followed by all SNPs, then putatively adaptive SNPs (Table 1). For Rocky Top only, observed heterozygosity was greatest with putatively adaptive SNPs. When private alleles were quantified for all sites, Rocky Top had 12 loci and YCC had 1 locus with private alleles in 2016 (Table 1). In 2020, only Rocky Top had seven loci with private alleles. Across all years, all private alleles were associated with putatively adaptive SNPs.

We identified a significant pattern of IBD for both years when calculated with all SNPs and neutral SNPs (2016 all SNPs, r = 0.224, *p*-value = 0.034; 2016 neutral SNPs, r = 0.339, *p*-value = 0.007; 2020 all SNPs, r = 0.361, *p*-value = 0.012; 2020 neutral SNPs, r = 0.371, *p*-value = 0.007). When calculated with adaptive SNPs, IBD was significant for 2020 (r = 0.361, *p*-value = 0.013) but not for 2016 (r = 0.0458, *p*-value = 0.34; Appendix A).

### 3.2. Evolutionarily Significant Units

The PCA performed with all SNPs identified three main groups: a western group, an eastern group, and only Rocky Top (Figure 2 and Appendix A). These groupings are also identified in STRUCTURE at *K* = 3, though the best supported *K* for STRUCTURE is *K* = 2 based on the Evanno method and *K* = 8 with the likelihood method (Appendix A). Due to Rocky Top’s distinctiveness in the PCA, its large number of private alleles, and that it is the first site to differentiate at *K* = 3, we have identified three ESUs: (1) a West ESU, (2) an East ESU, and (3) an ESU with only Rocky Top. The West ESU includes Summit Gulch (SG), Cap Gun/Tree Farm (CT), Steve’s Creek/Squirrel Valley/Manor (SS), Smith Mountain (SM), YCC (YC), Huckleberry (HU), Fawn Creek (FC), Cold Springs West (CW), Cold Springs East (CE), and North Hornet (NH). The East ESU consists of Lower Butter (LB), Slaughter Gulch (SL), Lost Valley (LV), Price Valley (PV), Tamarack (TA), Mud Creek (MC), and Hot Springs Road (HR). Each ESU has its own private alleles across 6 neutral and 5 putatively adaptive SNPs: 4 alleles for the West ESU, 5 alleles for the East ESU, and 2 alleles for the Rocky Top ESU (Appendix A).

### 3.3. Management Units

To delineate management units across population groups, we combined multiple analyses of demographic connectivity. We found significant levels of differentiation from pairwise *F_ST_* estimates by site (Figure 3, Appendix A). All pairwise *F_ST_* estimates were significantly different from zero in 2016, and all but two estimates in 2020 showed significant pairwise *F_ST_*. The majority of pairs of sites had *F_ST_* greater than 0.1 in 2016 and 2020, 71% and 78%, respectively (Appendix A). We detected strong genetic differentiation among populations; 32% (in 2016) and 52% (in 2020) of *F_ST_* estimates were greater than Wright’s [65] 0.15 threshold for strong genetic differentiation. For 2016, the mean and median *F_ST_* across all sites was 0.137 and 0.131 (range: 0.028–0.201), respectively. For 2020, the mean and median *F_ST_* across all sites was 0.159 and 0.156 (range: 0.007–0.331), respectively.

In the network analyses, we found that all sites were connected to at least one other site, and each year, all sites were connected to one another (Figure 4, Table 1). For 2016, we found a mean degree of 6 edges (range: 4–10) and a mean betweenness of 14 (range: 0–30) (Table 1). For 2020, we found a mean degree of 5 edges (range: 2–10) and a mean betweenness of 16.6 (range: 0–40).

The PCA revealed very similar genetic structure as ESUs: a western and an eastern grouping (Figure 3), where western sites were more spread across PC1 and sites on the periphery of the ranges are also distinct (Hot Springs Road and North Hornet). Hot Springs Road, North Hornet, Fawn Creek, Cold Springs East, and Smith Mountain differentiate along PC3 and PC4 (Appendix A).

The best supported value of *K* from STRUCTURE was *K* = 2 using the Evanno method and *K* = 9 using the likelihood method (Appendix A). The fine-scale structure supported by PCA and pairwise *F_ST_* align with *K* = 9 in STRUCTURE (Figure 3 and Appendix A). Therefore, we have delineated 9 MUs: (1) West MU of Summit Gulch, Cap Gun/Tree Farm, Steve’s Creek/Squirrel Valley/Manor, and Huckleberry, (2) Smith Mountain, (3) YCC, (4) Fawn Creek, (5) Cold Springs MU of Cold Springs West and Cold Springs East, (6) Rocky Top, (7) North Hornet, (8) East MU of Lower Butter, Slaughter Gulch, Lost Valley, Price Valley, Tamarack, and Mud Creek, and (9) Hot Springs Road.

With NeEstimator, we found that *N_e_* was variable among MUs (mean 38.16, range: 2.3–220.9) and was influenced by how many sites were within an MU (Table 2). The majority of MUs had an *N_e_* less than 50 individuals in 2016 and 2020, 67% (4/6 MUs) and 78% (7/9 MUs), respectively.

### 3.4. Adaptive Units

With adaptive SNPs, the PCA results agree with the STRUCTURE analyses (Figure 5). With STRUCTURE, *K* = 3 (with a subpeak at *K* = 5) is best supported using the Evanno method and *K* = 5 is supported with the likelihood method (Appendix A). There were no differences in the number of private alleles among *K*s 3, 4, or 5 (Appendix A). Moreover, all private alleles were represented at *K* = 3 that were also present at *K*s 4 and 5. Therefore, we have designated three AUs: (1) West AU, (2) East AU, and (3) an AU with just Rocky Top. The only difference between the AUs and the ESUs is that Smith Mountain is in the East AU, while it is in the West ESU.

### 3.5. The Distant Population of Round Valley

With only a single sample, we did not have enough data to fully include the isolated Round Valley population in our conservation unit analysis. The sample from Round Valley falls more closely with the west than the east group in PCAs with all, neutral, and adaptive SNPs (Appendix A). In the PCAs with all and neutral SNPs, Round Valley aligns with North Hornet, which is the geographically closest site to Round Valley. Round Valley is the most distinct along PC3 with only adaptive SNPs. Round Valley has no private alleles compared to the other sites.

## 4. Discussion

Through the application of a GT-seq panel in NIDGS [48], we were able to assess neutral and adaptive genetic variation to delineate conservation units, assess recovery goals, and help guide future management actions. Specifically, we delineated 3 ESUs, 9 MUs, and 3 AUs. The West ESU has 7 MUs, the East ESU has 2 MUs, and the Rocky Top ESU has 1 MU. The ESUs and AUs align with each other except for the assignment of one site. Across sites and conservation units, we quantified genetic variation, genetic structure, connectivity, and *N_e_* to address species recovery objectives to gauge the effects of previous management actions and guide future decision making.

### 4.1. Evolutionarily Significant Units

We found support for reproductive isolation in terms of the Waples [25] definition of ESUs. We detected clear separation among the west and east group in STRUCTURE results, with little to no admixture within individuals or sites (Figure 2 and Appendix A). This is also mirrored in the PCA results, where no individuals from either the west or the east fall out in the opposite grouping (Figure 2 and Appendix A). The division of a west and east group is supported by past NIDGS genetics work with mitochondrial DNA, nuclear microsatellites, and SNPs through PCA, STRUCTURE, and population graph analyses [42,43,47,66]. Broad scale population structure is likely most influenced by topographical features that serve as dispersal barriers, as west and east sites are separated by a mountain range [66]. Movement of NIDGS is less likely to occur over mountain ridges and would follow contour lines or rivers when possible [67].

Our ESU results support the Waples [25] definition of ESUs representing evolutionary distinctiveness. Private alleles (i.e., unique genetic variants) are found within each ESU (Appendix A). These alleles occur at both neutral and adaptive SNPs, which highlights that both genetic drift and selection may be contributing to divergence among the ESUs. Further, Rocky Top has the majority of private alleles when estimated per site, which are all associated with adaptive SNPs (Table 1). Gene flow may be limited to/from Rocky Top, and the unique habitat features of the site may have been selected for genetic variants unique to Rocky Top. For example, the soil is shallower and rockier with more sagebrush (*Artemisia* spp.) and less grass than other sites. Rocky Top is one of the higher elevation sites in our study (~1700 m). Phenological differences have been documented to vary by elevation across the NIDGS range, with squirrels at higher elevations immerging into and emerging from hibernation later with consequently shorter active seasons than squirrels at lower elevations [35]. Additionally, while we did not find any private alleles for the distant population Round Valley, we believe that further sampling of Round Valley is needed to assess its potential status as an ESU.

### 4.2. Management Units

Connectivity among occupied sites has been explored by previous NIDGS genetic studies, which have revealed patterns of fine-scale differentiation. Based on population graph analysis using microsatellite loci, the western sites had higher levels of connectivity with each other than sites in the east [66]. Connectivity within the western group was also found to be higher in previous STRUCTURE analyses, with sites sharing the same ancestral group at higher levels of *K* than the east sites [42,67]. However, assignment tests with microsatellites revealed a low level of exchange (9.38–23.08% mis-assignment) between sites overall, with sites in the east having slightly higher levels of mis-assignment than to sites in the west [43,67]. Fine-scale population structure is also supported by low to moderate levels of pairwise *F_ST_* reported across multiple studies [36,43,47]. The higher levels of differentiation in the east may be a consequence of loss of suitable habitat and dispersal corridors which has led to population isolation [36,38].

The levels of connectivity that affect genetic processes of drift and selection are different from the levels of connectivity that affect demographics. Genomic data alone cannot quantify demographic independence given that low rates of migration can maintain genetic similarity without causing significant demographic interdependence among populations [68]. For example, in our study we identified substantially more MUs than ESUs, reflecting lower levels of migration which create shared evolutionary history but not demographic connection. Given the fragmented habitat and limited dispersal of NIDGS [66], our identification of nine MUs within a relatively small area for this species is consistent with our previous understanding of these populations. In a previous gravity model intended to analyze functional connectivity for the NIDGS, site productivity and topography had the most effect on gene flow, suggesting these variables likely contribute to the fine-scale population structure we identified [66]. However, variables describing land cover, interspecific competition, and human disturbance were not found to impact functional connectivity [66]. To further our understanding of connectivity and demographic independence across the entire NIDGS range, further sampling at Round Valley is needed to assess its own classification as an MU given its geographic isolation.

The role of IBD in delineating MUs is also important to consider, as patterns of genetic differences are often correlated with geographic distance. Isolation by distance is fairly prominent across taxa and is not surprising for a small terrestrial mammal [69], but its significance here emphasizes that straight distance hinders connectivity among populations. An IBD relationship in NIDGS has been reported in some [42,47,66] but not all [43] prior studies. Each prior study examined a different subset of sites, and the inclusion or exclusion of different sites are important to consider in estimating the effect of geographic distance on allele frequencies. With the inclusion of samples from the peripheral sites, North Hornet and Hot Springs Road, the current study has a relatively larger geographic extent than previous studies.

### 4.3. Adaptive Units

Putatively adaptive SNPs for the GT-seq panel were originally identified with RADseq [42], with most of the sites in our study included in the RADseq SNP genotyping. That study used minimally invasive sampling and produced a final dataset of fewer than 4000 SNP loci, so there may be adaptive variation not included in the GT-seq panel. In Barbosa et al. [42], elevation appeared to be the main driver of adaptive differentiation between sites, and putatively adaptive SNPs were associated with environmental variables such as slopes, ridges, and peaks, as well as soil particle size. Further, the three populations that were the most distinct in the partial redundancy analysis (pRDA), Rocky Top, Lower Butter, and Tamarack, also showed separation in our PC axes 3 and 4 (Appendix A).

We identified five putatively adaptive SNPs that have private alleles among the three AUs. Gene ontology enrichment analyses, however, found no evidence for enrichment of specific molecular functions or biological processes linked with the putatively adaptive SNPs [42]. The only difference between the three ESUs and the three AUs is that Smith Mountain is grouped in the West ESU but within the East AU. In both analyses, this site showed evidence of shared ancestry with both the East and West groups. We only had 10 samples from Smith Mountain, and it would be valuable to continue sampling NIDGS at this site to understand local adaptation and connectivity of this population.

### 4.4. Management and Conservation Implications

We have more clearly defined the NIDGS metapopulation structure by evaluating genetic differentiation and connectivity. To be defined as a metapopulation, habitat patches/sites must represent independent breeding populations that are nonetheless connected to some extent to allow for recolonization following local extinction [70]. We found strong evidence that NIDGS form a loosely connected meta-population with strong genetic differentiation. However, despite the large degree of genetic differentiation between our sub-populations, they nonetheless represent a metapopulation rather than fully distinct populations. In a functioning metapopulation, patches/sites are connected to at least one other patch/site via gene flow [60]. In our population graph analysis, we found that all sites were connected. We did not include Round Valley in the Population Graph analysis, but it is likely isolated. Round Valley is ~15 km south of Lake Cascade and sits between two mountain ridges, which, as previous analyses suggest [66], are barriers to dispersal.

According to the species recovery plan, the NIDGS may be considered for delisting when ten subpopulations maintain an average *N_e_* of greater than 500 individuals for five consecutive years. From our estimates, no MU, and therefore no single subpopulation, has reached this recovery threshold. Seven out of nine MUs have an *N_e_* under 50 individuals, which is proposed to be the minimum *N_e_* necessary for preventing short-term inbreeding effects (but see [71,72]). However, the high *N_e_* in the West MU relative to the other MUs may suggest that it could be an important source population or buffer against population fluctuations in other parts of the species’ range. The West MU currently has the most samples available for analyses, and increased sampling within the other MUs would provide more accurate *N_e_* estimates for all MUs. Given the current fragmented habitat for this species, the recovery goal of *N_e_* > 500 across ten subpopulations may be difficult to achieve.

Population genetic metrics and conservation unit delineation assessed from genomic data can guide management decisions to preserve both short- and long-term species viability and persistence. In the short-term, priority is best placed on MUs with low levels of genetic diversity or *N_e_*, or those where *N_e_* is declining, because management units with low genetic diversity and *N_e_* are at higher risk of reduced fitness and loss of evolutionary potential [73]. Conserving small MUs is also critical to prevent against local extinction due to ecological or stochastic processes, such as predation [74], interspecific competition [75], and disease [44,45]. For NIDGS, conservation efforts may be most effective to focus on the three MUs (West, YCC, and Rocky Top) for which we detected a decrease in diversity and *N_e_* between 2016 and 2020, and two MUs (Smith Mountain and North Hornet) that had the lowest observed heterozygosity.

Maintaining connectivity within ESUs is more critical than maintaining connectivity between ESUs. To maintain connectivity within ESUs, it is most important to maintain dispersal corridors that link populations with small *N_e_* to reduce risks associated with genetic drift and inbreeding. Previous modeling suggests that prescribed burns would improve NIDGS habitat by reducing conifer density, increasing herbaceous community cover, and creating an open understory [39], but a recent experimental study that tested these predictions did not find evidence that using prescribed burning is effective [76].

In the long-term, it is important to maintain adaptive variation among ESUs/AUs, because it determines long-term species viability, as well as the potential for species distribution or population size to increase [15]. It is vital to conserve species-wide adaptive diversity, especially in species composed of small, isolated subpopulations, to maintain evolutionary potential and counteract the influence of genetic drift [73]. Therefore, Rocky Top is valuable as an isolated ESU/AU and more explicit studies on the Rocky Top population may help identify its unique attributes. Rocky Top shows signs of high adaptive genetic variation, which may harbor capacity to respond or adapt to environmental change. For example, NIDGS hibernation behavior is sensitive to changes in snowfall, ambient temperature, and food availability, and altered hibernation emergence will likely impact survival [35]. Habitat features that reduce the risk of predation, during both the active season and the hibernation season, are important to identify as well as the habitat features that buffer NIDGS from harsh weather conditions. For instance, snowpack during hibernation may be important for both reducing predation and ameliorating suboptimal temperatures [35]. Future climate projections for west-central Idaho predict more rain and less snow [77]; therefore, NIDGS may experience a suite of selection pressures that have not been present in their evolutionary history. As such, maintaining genetic variation will be especially critical for the species’ persistence as the climate changes. Additionally, for management actions like translocations or genetic rescue, only moving individuals from within ESUs/AUs would minimize the risk of outbreeding depression [78,79].

The continuation of genetic monitoring into the future is critical to evaluate the effects of management action or stochastic events. The GT-seq panel developed for the NIDGS [48] is a useful tool for monitoring allele frequencies and gene diversity at both neutral and putatively adaptive SNPs as more samples are collected over time. Estimates of gene flow between sites or detections of migration can identify potential dispersal corridors that can be maintained by habitat protection or restoration. Checking for losses of heterozygosity or decreases of *N_e_* will provide insights on the effects of inbreeding. Genetic monitoring in NIDGS that includes estimates of genetic drift and inbreeding, as well as adaptive variation, will best inform estimates of species persistence. The inclusion of future samples from previously sampled sites or new sites will validate or adjust conservation unit delineation and will help with the prioritization of resource allocation to conservation units. The inclusion of other data types, like ecological or behavioral data, would be valuable to collect and incorporate to confirm or re-evaluate conservation units. Overall, genomic data provides critical information that can directly assess recovery goals and actions for this threatened species.

## Figures and Tables

**Figure 1 genes-16-00694-f001:**
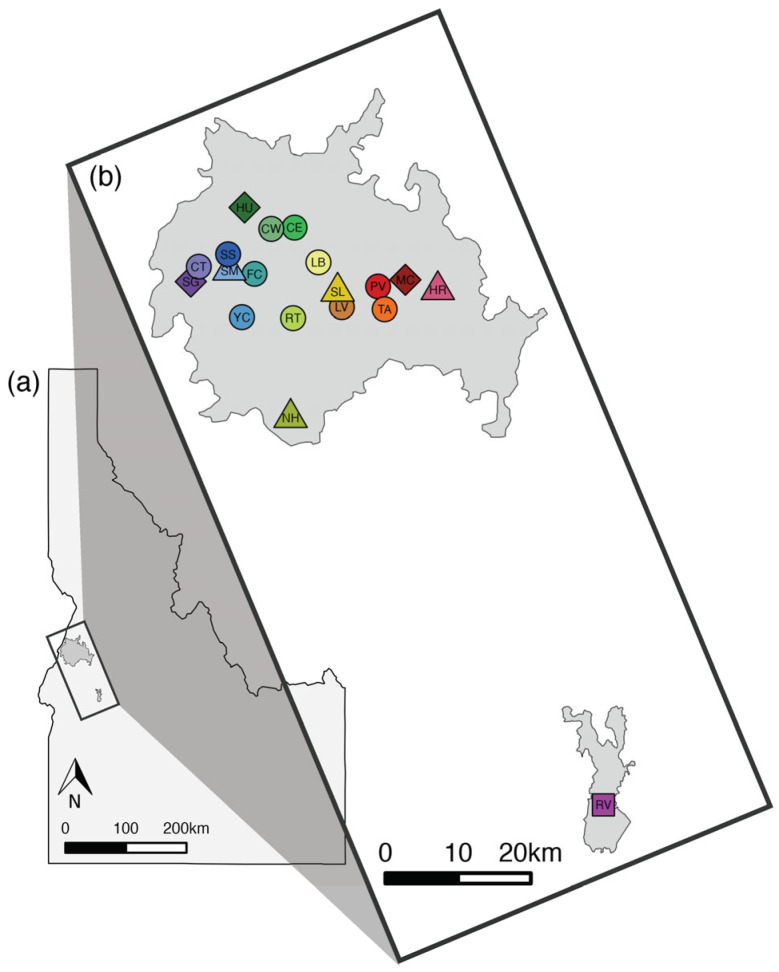
Location of sampling sites within the NIDGS range in central Idaho, USA, including the years each site was sampled. (**a**) Outline of Idaho with range of NIDGS in dark gray. (**b**) Detail of NIDGS range with sampling sites denoted by color (which correspond to PCAs and population graphs in subsequent figures); full site names are in Table 1. Shape represents collection year: diamond = 2016, triangle = 2020, circle = 2016 and 2020, square = Round Valley in 2022. NIDGS range obtained from https://ecos.fws.gov/ecp/species/2982 (accessed on 6 June 2024).

**Figure 2 genes-16-00694-f002:**
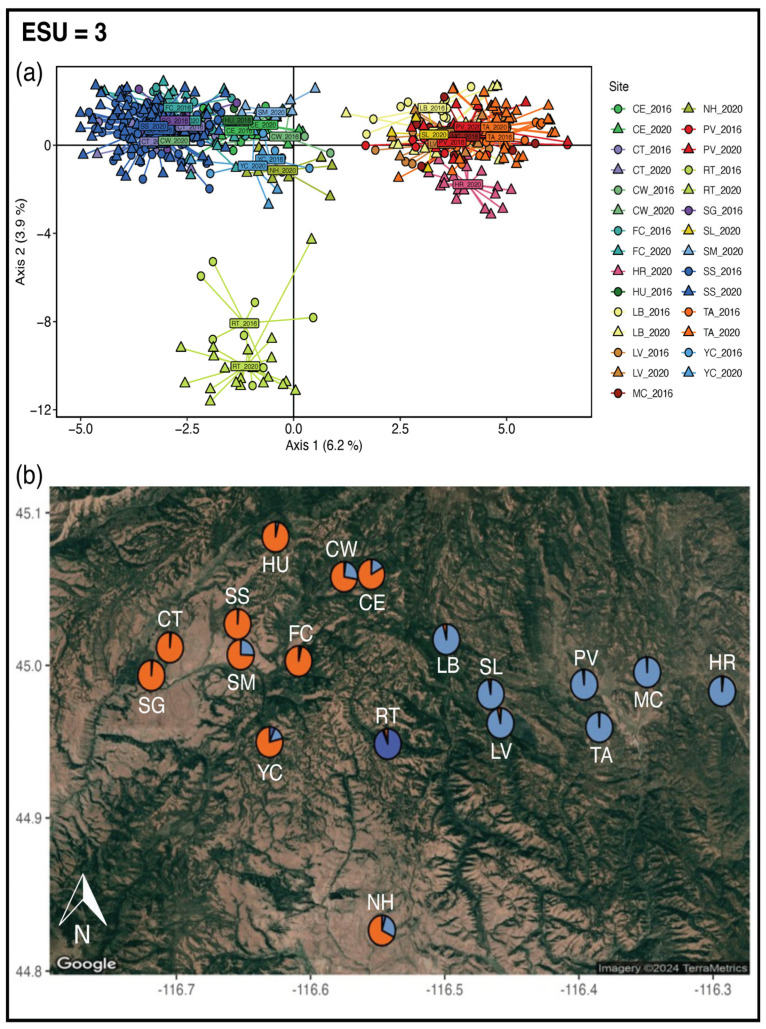
Outputs from PCA and STRUCTURE used to delineate three ESUs. (**a**) PCA with all SNPs; circles represent samples from 2016, and triangles represent samples from 2020. Colors correspond to site labels in Figure 1. (**b**) Ancestry values from STRUCTURE are given as pie charts plotted on the NIDGS range. Each color represents a distinct genetic cluster.

**Figure 3 genes-16-00694-f003:**
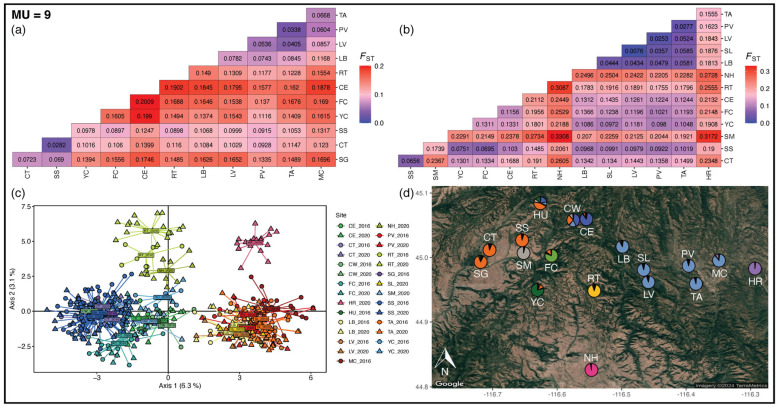
Outputs from pairwise *F_ST_*, PCA, and STRUCTURE used to delineate nine MUs for the NIDGS. (**a**) Pairwise *F_ST_* heatmap with neutral SNPs for 2016 samples. Blue represents *F_ST_* values of 0 and red represents *F_ST_* values of 0.2. (**b**) Pairwise *F_ST_* heatmap with neutral SNPs for 2020 samples. Blue represents *F_ST_* values of 0 and red represents *F_ST_* values of 0.3. (**c**) PCA with neutral SNPs, circles represent samples from 2016, and triangles represent samples from 2020. Colors correspond to site labels in Figure 1. (**d**) Ancestry values from STRUCTURE with neutral SNPs are given as pie charts plotted on the NIDGS range. Each color represents a distinct genetic cluster.

**Figure 4 genes-16-00694-f004:**
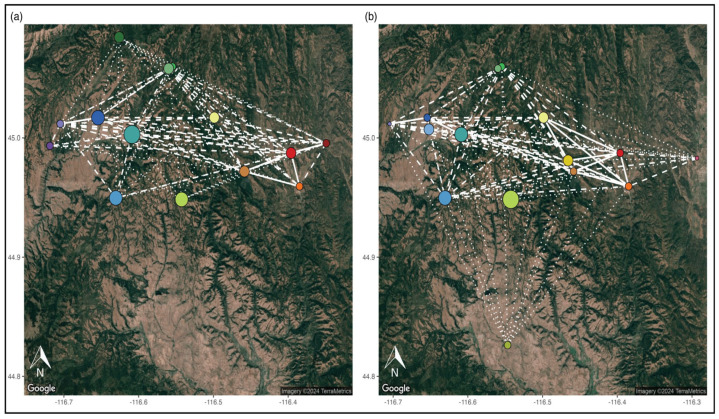
Saturated spatial population graph illustrating the gene flow between NIDGS sites for (**a**) 2016 and (**b**) 2020. Edges shown are flow predictions (flow = 1 − Nei’s *D*): dotted = 0.85–0.90; dashed = 0.90–0.95; solid = 0.95–1.0. The sizes of the nodes are scaled to measure degree (total number of edges connected to a particular node), and the color represents sites and is the same as Figure 1 and all the PCAs.

**Figure 5 genes-16-00694-f005:**
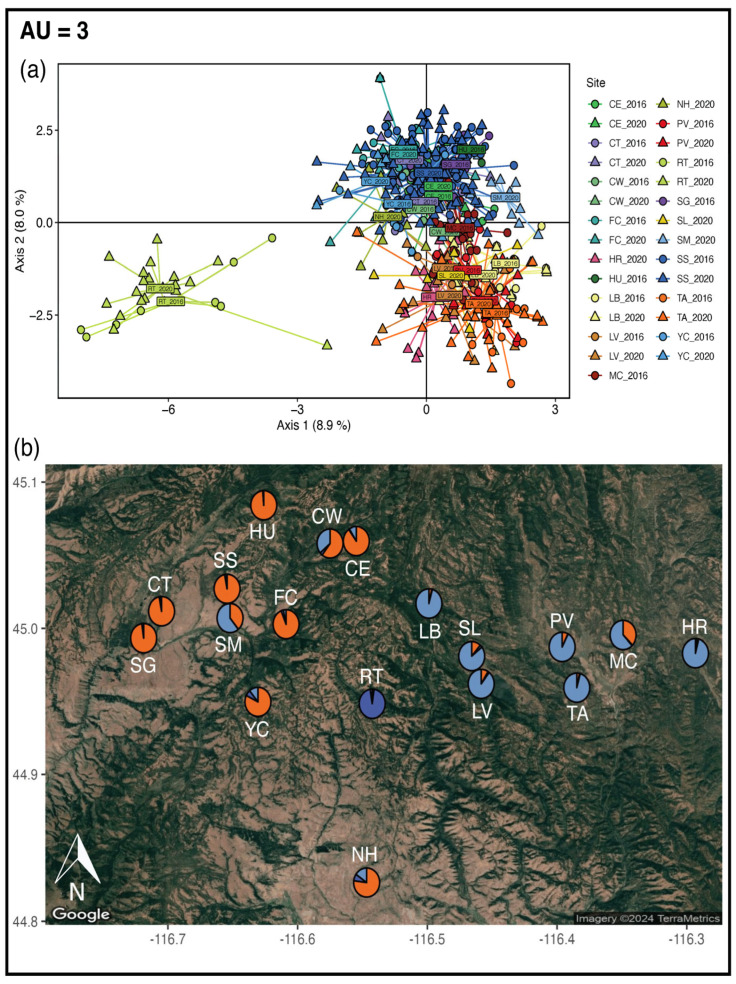
Outputs from PCA and STRUCTURE used to delineate three AUs for the NIDGS. (**a**) PCA with putatively adaptive SNPs, circles represent samples from 2016, and triangles represent samples from 2020. Colors correspond to site labels in Figure 1. (**b**) Ancestry values from STRUCTURE with adaptive SNPs are given as pie charts plotted on the NIDGS range. Each color represents a distinct genetic cluster.

**Table 1 genes-16-00694-t001:** Site level measures for 2016 and 2020 for the NIDGS. *N* is the sample size. All private alleles were observed at adaptive SNPs. Degree and betweenness values were estimated using Population Graphs [56].

Site Name	Site Abbr.	Year	*N*	Observed Heterozygosity			Expected Heterozygosity			Private Alleles	Degree	Betweeness
				All SNPs	Neutral SNPs	Adaptive SNPs	All SNPs	Neutral SNPs	Adaptive SNPs			
Cold Springs East	CE	2016	10	0.270	0.308	0.174	0.254	0.285	0.173	0	4	14
		2020	12	0.292	0.323	0.210	0.275	0.308	0.192	0	4	12
Cap Gun/Tree Valley	CT	2016	6	0.291	0.325	0.201	0.303	0.345	0.194	0	4	5
		2020	8	0.274	0.308	0.186	0.278	0.314	0.187	0	4	9
Cold Springs West	CW	2016	4	0.291	0.311	0.238	0.302	0.319	0.261	0	6	14
		2020	1	0.305	0.358	0.171	NA	NA	NA	0	2	0
Fawn Creek	FC	2016	15	0.265	0.287	0.208	0.258	0.279	0.203	0	10	22
		2020	19	0.297	0.317	0.245	0.283	0.302	0.233	0	8	29
Hot Springs Road	HR	2020	19	0.294	0.315	0.240	0.280	0.299	0.229	0	2	0
Huckleberry	HU	2016	2	0.309	0.363	0.174	0.208	0.242	0.119	0	6	0
Lower Butter	LB	2016	12	0.334	0.370	0.240	0.291	0.320	0.213	0	6	14
		2020	15	0.327	0.351	0.266	0.314	0.334	0.262	0	6	14
Lost Valley	LV	2016	10	0.298	0.332	0.211	0.302	0.337	0.213	0	6	30
		2020	16	0.308	0.333	0.244	0.311	0.339	0.238	0	4	33
Mud Creek	MC	2016	22	0.308	0.356	0.179	0.301	0.345	0.185	0	4	0
North Hornet	NH	2020	10	0.229	0.243	0.192	0.240	0.264	0.177	0	4	0
Price Valley	PV	2016	10	0.318	0.350	0.236	0.312	0.346	0.222	0	6	22
		2020	14	0.326	0.358	0.245	0.315	0.350	0.224	0	4	24
Rocky Top	RT	2016	8	0.334	0.330	0.343	0.304	0.295	0.328	12	8	22
		2020	20	0.314	0.298	0.353	0.278	0.269	0.303	7	10	34
Summit Gulch	SG	2016	7	0.266	0.320	0.125	0.275	0.323	0.151	0	4	1
Slaughter Gulch	SL	2020	7	0.308	0.343	0.217	0.303	0.331	0.230	0	6	40
Smith Mountain	SM	2020	10	0.241	0.264	0.183	0.238	0.263	0.174	0	6	24
Steve’s Creek/Squirrel Valley/Manor	SS	2016	77	0.309	0.351	0.199	0.311	0.353	0.202	0	8	14
		2020	74	0.302	0.343	0.193	0.308	0.351	0.195	0	4	13
Tamarack	TA	2016	12	0.344	0.360	0.302	0.334	0.355	0.277	0	4	0
		2020	39	0.336	0.362	0.270	0.333	0.358	0.265	0	4	13
YCC	YC	2016	6	0.310	0.328	0.262	0.282	0.300	0.239	1	8	6
		2020	6	0.295	0.309	0.261	0.300	0.316	0.258	0	8	4

**Table 2 genes-16-00694-t002:** Effective population size for each MU estimated from NeEstimator v. 2 [64] using the linkage disequilibrium method. *N* is sample size; *H_O_* is observed heterozygosity calculated with all loci and only neutral loci; lower and upper 95% confidence intervals (CI) for each MU.

Management Unit	Year	*N*	*H_O_*—All SNPs	*H_O_*—Neutral SNPs	*N_e_*	Lower CI	Upper CI
West	2016	92	0.304	0.347	220.9	147.9	403.7
	2020	82	0.299	0.34	120.8	69.8	305.6
SM	2020	10	0.241	0.264	18.7	8.8	92.5
YC	2016	6	0.31	0.328	18.8	5	inf
	2020	6	0.295	0.309	2.3	0.7	inf
FC	2016	15	0.265	0.287	6.7	2.7	16.2
	2020	19	0.297	0.317	11	5.1	26.9
Cold Springs	2016	14	0.276	0.306	14.5	9.2	26.2
	2020	13	0.293	0.326	2.7	1.3	16.1
RT	2016	8	0.334	0.33	24.6	9.8	inf
	2020	20	0.314	0.298	7	3.3	11.6
NH	2020	10	0.229	0.243	11.5	5.3	34.8
East	2016	66	0.32	0.355	50.7	36.9	74.1
	2020	91	0.326	0.353	50.6	35.8	75.9
HR	2020	19	0.294	0.315	11.6	5.8	27

## Data Availability

Genotyping pipelines, custom scripts, and the GT-seq SNP dataset in .csv format are available on GitHub (https://github.com/mjgarrett/NIDGS_scripts, accessed on 4 June 2024). VCF files from the RADseq data are available on Dryad (https://doi.org/10.5061/dryad.sj3tx965c, accessed on 4 June 2024).

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
