# Peer review of "Genetic Variation and Metapopulation Structure Inform Recovery Goals in a Threatened Species"

_genes, 2025, doi:10.3390/genes16060694_

Round 1
Reviewer 1 Report
Comments and Suggestions for Authors
The manuscript describes population genetics analyses of Northern Idaho ground squirrels based on a GT-seq methodology. The topic is very interesting and also important from a conservation point of view. The structure and length of the manuscript are deliberate, although some parts might be a little bit shorter. The text is clear and concise. The presentation of the results and the discussion are fairly detailed and straightforward. I have only some minor suggestions.
In section 2.1, the sampling locations could be described in more detail. Although the abbreviations are resolved later in the manuscript, I would find it helpful if the locations were listed here.
The title of section 2.4 should read “Management Units”.
The site RT is listed in Table 1 as “Rockytop” but mentioned in the text as “Rocky Top”. You should be consistent with names.
Reviewer 2 Report
Comments and Suggestions for Authors
This study evaluated the genetic diversity, structure, connectivity, and effective population size of Urocitellus brunneus and applied Genotyping-in-Thousands by Sequencing (GT-seq) panels with the aim of species recovery.The results revealed the presence of three evolutionarily significant units, nine management units, and three adaptive units capturing adaptive differentiation, and indicated that the population sizes within each management unit have not yet reached recovery targets. This study demonstrates the utility of GT-seq and provides guidelines for the conservation management of Urocitellus brunneus.
L160: QiagenATL buffer (company name)
L209-214: Are you using other software in the process from Structure analysis to plot drawing? (It is difficult to see the PLOT in STRUCTURE, although it seems to be one of the output formats. If the opinions of other reviewers and editors are similar, please change it.)
Methods: It would be helpful if all software had version information.
L232: Ne is also more reliable data when calculated by multiple methods.
P254: Selection of SNPs from RAD-seq data. Are you selecting SNPs considering Hardy-Weinberg equilibrium, linkage disequilibrium?
Microsatellites are excellent for assessing the genetic diversity of local populations. Why not use microsatellites? We have analyzed whole genomes in your previous studies and I believe that microsatellite identification is feasible and an option. With 254 SNPs, I don't see much advantage over microsatellite analysis. You should mention a little more in the introduction or discussion that GT-seq is more useful for the analysis than other methods.
There are examples of previous studies using microsatellites, so if you have done comparisons between populations, please compare the results of your studies.
P13: It is good to use multiple methods to analyze differences between populations; PCA has low first and second principal components and clusters are somewhat unclear (Figure 5) In the analysis of AU, why not show Structure analysis and phylogenetic trees in the text?
Introduction or Discussion:
It would be nice to denote the longevity and generation length of NIDGS. It is unclear how long four years is equivalent to for NIDGS. The L496-498 discussion also helps to evaluate the impact of 4 years on the population
L388: Please describe the ecology of NIDGS, especially the migration of females and males.
